# High Performance GaN-Based Ultraviolet Photodetector via Te/Metal Electrodes

**DOI:** 10.3390/ma16134569

**Published:** 2023-06-24

**Authors:** Sheng Lin, Tingjun Lin, Wenliang Wang, Chao Liu, Yao Ding

**Affiliations:** 1School of Materials Science and Engineering, Wuhan University of Technology, Wuhan 430070, China; 2Department of Electronic Materials, School of Materials Science and Engineering, South China University of Technology, Guangzhou 510640, China; 3State Key Laboratory of Crystal Materials, School of Microelectronics, Institute of Novel Semiconductors, Shandong Technology Center of Nanodevices and Integration, Shandong University, Jinan 250100, China; chao.liu@sdu.edu.cn

**Keywords:** Te-enhanced, GaN, straddling band, ultraviolet photodetectors

## Abstract

Photodetectors (PDs) based on two-dimensional (2D) materials have promising applications in modern electronics and optoelectronics. However, due to the intralayer recombination of the photogenerated carriers and the inevitable surface trapping stages of the constituent layers, the PDs based on 2D materials usually suffer from low responsivity and poor response speed. In this work, a distinguished GaN-based photodetector is constructed on a sapphire substrate with Te/metal electrodes. Due to the metal-like properties of tellurium, the band bending at the interface between Te and GaN generates an inherent electric field, which greatly reduces the carrier transport barrier and promotes the photoresponse of GaN. This Te-enhanced GaN-based PD show a promising responsivity of 4951 mA/W, detectivity of 1.79 × 10^14^ Jones, and an external quantum efficiency of 169%. In addition, owing to the collection efficiency of carriers by this Te–GaN interface, the response time is greatly decreased compared with pure GaN PDs. This high performance can be attributed to the fact that Te reduces the contact resistance of the metal electrode Au/Ti to GaN, forming an ohmic-like contact and promoting the photoresponse of GaN. This work greatly extends the application potential of GaN in the field of high-performance photodetectors and puts forward a new way of developing high performance photodetectors.

## 1. Introduction

In recent years, photodetectors (PDs) have experienced a vigorous development. Among the variety of PDs, ultraviolet photodetectors (UV PDs) can work in the UV bands and are of great importance in applications of the military and industrial fields, such as video imaging [1,2], night-vision equipment [3], missile warning [4], etc. Among all the newly emergent materials for PDs, two-dimensional (2D) materials have crucial and promising applications in the fields of catalysis [5], energy storage [6], and sensors [7], due to their unique physicochemical properties and surface electron-limited domains. Their unique properties such as the atomic layer thickness, van der Waals force superposition, and fewer surface chemical suspension bonds have led to the extensive study of 2D materials in various PD fields in different spectral ranges [8,9,10,11,12]. For example, recent studies on perovskite thin-film materials have found that their large specific surface area and tunable surface activity can provide superior charge transport paths, allowing 2D perovskite thin-film materials to show tunable optical responses and short response times as photodetectors [13,14,15]. At present, there are two main directions for improving the performance of photodetectors. One is to develop different device structures, such as metal–semiconductor–metal (MSM) PDs [16,17], Schottky barrier PDs [18,19], etc. The former utilizes Schottky contact between the metal and semiconductor to generate an electric field so as to effectively absorb photons. However, when the illumination becomes stronger, the large amount of carriers shield the electric field and reduce the transmission of carriers between the two Schottky contacts. The other approach is to search the materials with high photoresponse performance, such as AlGaN [20], PtSe_2_ [21], and their heterogeneous structures [22,23]. This method utilizes the optical response characteristics of the semiconductor itself to reduce the carrier transmission barrier by forming the type Ⅱ heterojunctions and generating band bending. However, due to the limited band gap of materials, the wide response range and high response rate of this kind of photodetector cannot be achieved at the same time.

Two-dimensional gallium nitride (GaN) has recently received a lot of attention for its ability to provide high performance UV PDs [24,25]. Spontaneous polarization in GaN films often results in slow response and poor responsiveness. Tellurium has excellent metal-like electrical conductivity and narrow band gap properties. Therefore, when tellurium is applied to a Gan-based PD, it can effectively provide a tunnel effect and inhibit photogenerated electron–hole pair recombination, which improves the conversion efficiency of PDs. On the other hand, the Te/metal electrodes can reduce the energy barrier between GaN and metal electrodes by generating band bending under illumination. This contributes to the GaN–Te/metal ohmic-like contact by reducing the contact resistant and promotes the photoresponse properties of GaN.

In this work, we synthesized the high-quality Te/GaN heterojunctions by firstly obtaining the Te nanowires (Te NWs) through a simple hydrothermal method and then spin coating on the GaN/sapphire substrate. Due to the difference in work function, Te and Au generate asymmetric bands, which promote carrier transmission. In addition, as proved by the theoretical calculations based on density functional theory (DFT) and the Kelvin probe force microscopy (KPFM), the large energy band mismatch between Te and GaN can offer a special band-to-band tunneling effect and suppress the dark current. In addition, due to the ultrahigh electronic conductivity of Te, it can effectively collect the photogenerated carriers in GaN, which helps to inhibit the recombination of photogenerated carriers in PDs. Benefiting from the above advantages of Te/metal electrodes, the photodetectors based on the Te/GaN heterojunctions have an ultralow dark current of 3.1 × 10^−7^ A and good photovoltaic effect. The Te-enhanced GaN photodetectors have a response rate and detection rate of 4951 mA/W and 1.79 × 10^14^ Jones, respectively, with a maximum EQE value of 169%, which exceeds most Gan-based photodetectors in the reported works. This work greatly develops the application potential of GaN in the field of high-performance photodetectors and puts forward a new way of developing high-performance photodetectors.

## 2. Materials and Methods

### 2.1. Preparation of the GaN Film on Sapphire

The 2 inch GaN-on-sapphire wafers were provided by Suzhou Nanowin Science and Technology Co., Ltd. in Suzhou, China. The GaN (0001) layers are 4.5 ± 0.5 μm thick, and have a carrier concentration of less than 5 × 10^17^ cm^−3^.

### 2.2. Preparation of the Te Nanowires

The synthesis of tellurium nanowires (Te NWs) is carried out by a typical hydrothermal method, as illustrated in Figure 1 [26,27]. First, 61.6 mg of Na_2_TeO_4_·2H_2_O and 43.2 mg of PVP are dissolved in deionized (DI) water and stirred for 30 min to form a homogeneous solution. After that, 2 mL mixed solution of ammonia and hydrazine hydrate with a volume ratio of 2:1 is added to obtain the solution with pH = 11.1. Finally, the obtained solution is transferred into the reactor for a hydrothermal reaction for 30 h at 180 °C. The products are centrifuged and washed with DI water and anhydrous ethanol several times to obtain the tellurium NWs. Typically, Te NWs with a thickness of ~90 nm and a length up to 30 μm can be obtained by this synthetic method.

### 2.3. Preparation of the Te/GaN Heterostructure

For the preparation of Te/GaN heterojunctions, Te NWs are dispersed in ethanol and then spin-coated onto the single crystalline GaN (0001) substrate with 300 rpm, 10 s and 2000 rpm, 30 s, respectively. The density of Te NWs on the substrate can be adjusted by the concentration of Te NWs in the dispersion.

### 2.4. Characterization of Te NWs and GaN Film

The morphology of the resulting nanowires was characterized by optical microscopy (50/100× objective, purchased from Sunny Optical Technology (Group) Co., Ltd. in Ningbo, China) The thickness of the 2D material was measured by atomic force microscopy (SmartSPM 1000, purchased from AIStar-Technology company, Novato, CA, USA). Raman testing was acquired by Raman spectroscopy system (Horiba LabRAM HR Evolution, purchased from HORIBA, Ltd. in Kyoto, Japan) under 532 nm laser excitation. X-ray diffractometer (D8 discover X-ray, purchased from Bruker company in Billerica, MA, USA) and transmission electron microscope (Talos F200S, purchased from Thermo Fisher Scientific in Waltham, MA, USA) were used to characterize the structure of nanowires. KPFM was tested on an atomic force microscope (SPM-9700HT, Shimadzu Co., Ltd. in Nagoya, Japan).

### 2.5. Fabrication of Te/GaN Photodetectors

The electrodes in the device, including the marker electrodes and the inner electrodes, are deposited with Ti and Au metals of 10 nm and 80 nm, respectively, using the thermal evaporation physical vapor deposition method. Among them, the marker electrodes are realized by UV lithography. First, a layer of photoresist is spin-coated on the substrate, and then the marker pattern channel is formed under UV irradiation with the aid of a mask plate, after which the marker electrode is deposited. The inner electrode is obtained using the EBL technique. First, a layer of photoresist is spin-coated on the substrate where the marker electrode is prepared, and then the NPGS system is programmed to control the trajectory of the electron gun and the stabilization current level, etc. A current of 400 pA is used to etch at a suitable location to obtain the inner electrode pattern channel, and then the electrode is deposited to obtain the inner electrodes.

### 2.6. Characterizations of the PDs and Photovoltaic Measurements

A power device analyzer (KEYSIGHT b1502a) was used to measure the electrical and optoelectronic properties of the resulting devices. During the test, the device is irradiated by laser with different wavelengths and power density. The intensities of the 254 nm, 365 nm, and 810 nm illuminants are 33 μW/cm^2^, 120.6 mW/cm^2^, and 14.7 mW/cm^2^, respectively. The distance of laser source and the PDs is 1 m.

### 2.7. DFT Calculations

All calculations in this work were performed on the VASP platform based on the DFT principle [28,29]. The exchange–correlation effects were described by the Perdew–Burke–Ernzerhof (PBE) functional within the generalized gradient approximation (GGA) method [30,31]. The core–valence interactions were accounted for by the projected augmented wave (PAW) method [32]. The energy cutoff for plane wave expansions was set to 450 eV. The Brillouin zone was sampled by using Monkhorst–Pack grid k-points 16 × 16 × 10 for GaN, 13 × 13 × 10 for Te, and 4 × 4 × 1 for GaN/Te. The structural optimization was completed for energy and force convergence set at 1.0 × 10^−6^ eV and 0.01 eV Å^−1^, respectively.

## 3. Results and Discussion

The crystal structures of the Te/GaN heterojunction are explored in Figure 2. For Te NWs, Te atoms are connected by strong covalent bonds within the chain to form a helical chain along the *c*-axis, while the chains are arranged parallel to each other and connected by weak van der Waals forces to form a hexagonal lattice structure [33]. The crystal structure of GaN is a fibrillated zincite structure with an AB stacking mode, the P63mc space group, which has alternating layers of closely packed (0001) Ga metal atoms and N atoms [34]. The Te/GaN heterojunction is formed by Te nanowire and GaN substrate with weak van der Waals interaction (Figure 2a). In order to obtain a clean interface of the heterojunction, GaN substrate is carefully cleaned by sonication in ethylene glycol and DI water before the preparation. Figure 2b shows the optical microscopy image of single Te nanowire, while the height plot of Te NW in Figure 2c indicates that it has a thickness of ~90 nm. Furthermore, Raman results on GaN and the Te/GaN heterojunction show that two characterization peaks of Te can be observed at 120 cm^−1^ (A_1_ band) and 140 cm^−1^ (E_2_ band) in the Te/GaN heterojunction (Figure 2d), consistent with previous reports [35]. Here, the A_1_ mode presents the chain extension mode of each atom vibration in the basal plane, while the E_2_ mode can be attributed to the bond stretching on the c-axis [35,36]. In addition, Raman spectra for both GaN and the heterojunction show two characteristic peaks at 564 cm^−1^ and 730 cm^−1^, corresponding to the E_2_ and A_1_ modes of GaN, respectively [37]. To further directly view the crystalline quality of the Te NWs in the heterojunction, transmission electron microscopy (TEM) analysis of the nanowire is performed. Figure 2e,f show the low- and high-resolution TEM images of the Te NWs, which shows that it is a three-dimensional (3D) structure with a trigonal space group of $D_{3}^{4}$ [38]. Figure 2f presents the Te nanowire with an interplanar spacing *d* = 0.59 nm, corresponding to the Te (0001) planes, while no obvious defects can be observed. Meanwhile, the selective area electron diffraction (SAED) patterns derived from the red dash circular area in Figure 2e (inset in Figure 2f) confirms the highly crystalline structure. The SAED patterns corresponding to the selected region show highly defined spots of Te (0001) and (11$\bar{2}$0). The above features indicate the Te nanowires in the heterojunctions are of high quality.

To deeply investigate the band alignment of the Te-enhanced electrodes on GaN photodetectors, theoretical studies based on density functional theory (DFT) are conducted on Te, GaN, and Te/GaN heterojunctions. Figure 3a,b shows the device schematic of the Te-enhanced GaN and bare GaN photodetectors. The corresponding photograph of the Te-enhanced GaN photodetector is shown in Figure 3c. For Te-enhanced GaN photodetectors, Te is the drain electrode and GaN is the source electrode during the test. The band alignment of the Te/GaN heterojunctions is analyzed before the photovoltaic test. First, the calculated energy band structures of Te and GaN are shown in Figure 3d,e (details in the Experimental Section). The results indicate that Te has a direct band gap of 0.17 eV with the Fermi energy level at 0.97 eV, while GaN has an indirect band gap of 1.75 eV with the Fermi energy level at 1.59 eV. This is consistent with the previous references [39,40]. To experimentally observe the contact mode of GaN/Te/Ti/Au, the electronic transportation properties of bare Te, bare GaN, and Te-enhanced GaN devices are performed in Figure 3f. From the IV curves, we can know that bare Te has excellent ohmic contacts with Ti/Au electrodes, while bare GaN has the typical Schottky contacts. In addition, the electronic conductivity of Te is much better than that of GaN and behaves almost as a semi-metallic material. Importantly, for Te-enhanced GaN devices, the resistance between GaN and Ti/Au electrodes is greatly reduced. To give a clear illustration from the band structure view, Figure 3g shows the energy band diagram of the Ti/Au-GaN-Ti/Au device. Since the GaN has the Schottky contact with the Ti/Au electrode, the built-in electric field generated by these two Schottky contacts has an opposite direction, causing the energy band at the interface to bend upward and form an electron transport potential barrier. However, when the electrode changes to the Te/GaN heterojunction, an internal electric field is formed at the Te/GaN interface. Since the Fermi energy level of Te is higher than that of GaN, a built-in electric field is formed pointing from Te to GaN [41], leading to the rearrangement of the energy bands on the contact side of GaN and Te to bend downward, and such an energy band distribution is favorable for carrier transport, as shown in Figure 3h.

In order to provide the solid experimental evidence for the above illustration of band alignment between Te/Ti/Au electrodes and GaN, surface potential difference (SPD) measured by KPFM has been used to determine the relative position of Femi levels in the heterojunction under dark and illumination. Figure 4a,b show KPFM diagrams of the Te–GaN interface in darkness and light, from which it can be seen that GaN has a slightly higher surface potential of 0.08 eV than Te in dark environments (Figure 4c). Thus, GaN has a higher Femi level, which causes the band to bend downward from GaN to Te and electrons tend to aggregate in Te, consistent with the theoretical explanation in Figure 3h. On the contrary, the surface potential of Te increases to be higher than that of GaN under illumination of 365 nm, as shown in Figure 4d. At the same time, the SPD increases significantly to ≈0.23 eV, which strongly proves that the photogenerated electron–hole pair has been separated and electrons flow from Te to GaN. Furthermore, because of the fact that Au tips are used in the test and the SPD value is negative, the surface potential of Au should be lower than that of GaN and Te, consistent with the illustration in Figure 3g.

Based on the above theoretical works about the band alignment of Te-enhanced GaN devices, micro-photodetectors (m-PD) constructed by GaN with Te/Ti/Au electrodes are fabricated by photolithograph and electron beam lithography (EBL) (details in Experimental section) to study the photovoltaic (PV) behavior of the Te-enhanced GaN PDs. Here, plenty of devices have been fabricated to obtain convincing results (Appendix A). For the test of PDs, the four-point probe measurement was conducted to obtain the PV behavior. To further investigate the photovoltaic performance of the Te-enhanced GaN photodetector, bias voltages of −5 to +5 V are applied under dark and illumination (365 nm), respectively. The asymmetric logarithmic I–V curves are plotted in Figure 5a, while the inset shows the linear I–V curve of the photodevice in dark. Figure 5a indicates that the Te-enhanced GaN photodetector shows an ultralow dark current of 3.1 × 10^−7^ A. Furthermore, the photoresponse characteristics of the Te-enhanced GaN photodetector at other wavelengths are measured under 254, 365, and 810 nm illumination, as shown in Figure 5b. It can be found that the photocurrent of GaN has a sharp increase under illumination, especially under 365 nm illumination. At −5V bias, the photocurrent increases from 0.8 mA at 254 nm to 30 mA at 365 nm. It should be noted that GaN should have no optical response at wavelengths larger than 365 nm [42,43,44]. Therefore, the photocurrent illuminated at 810 nm probably originated from the Te NWs, which are reported as well-performed near-infrared (NIR) photodetectors [45]. The short-circuit current (I_SC_) and open-circuit voltage (V_OC_) of the Te-enhanced GaN photodetector under 365 nm illumination are 1.02 μA and 0.04 V, respectively (Figure 5c). Here, the different behavior of Te-enhanced GaN in forward and reverse bias under dark and illumination can be explained in Figure 5d–f. When forward bias is applied, the direction of the applied electric field points from Te to GaN, and the Fermi energy level of GaN rises and the energy band bends downward, which allows the electrons to move freely from GaN to Te. In contrast, when reverse bias is applied, the direction of the applied electric field is opposite to that of the built-in electric field generated by the Te–GaN interface, thus, a steep potential barrier forms at the interface, preventing electrons in GaN from drifting into Te, which results in a relatively small current of about 10^−5^ A at forward bias. However, when the heterojunctions are illuminated at 365 nm, due to the fact that the photon energy of 365 nm is much higher than the band gap of Te (0.17 eV), plenty of photogenerated carriers are created. Therefore, electrons can easily overcome the steep energy barrier and jump to the conduction band of GaN under the action of external electric field (Figure 5f). At the same time, the band shift between the conduction band of Te and GaN is significant, which also promotes the injection of electrons. As a result, the spatial separation of the photogenerated electron–hole pairs is promoted (Figure 4b), and the drift current combined with the tunneling current leads to a significant increase in the photocurrent under reverse bias. By introducing the Te/Ti/Au electrodes, it can greatly solve the Fermi pinning effects in bare GaN PDs.

Then, the photovoltaic behavior of the Te-enhanced GaN PDs illuminated at 365 nm with different power intensity are studied in details. Figure 6a shows the I–V curves of the Te-enhance GaN PDs measured at dark and 365 nm with different power intensity varying from 16.44 mW/cm^2^ to 120.6 mW/cm^2^. From Figure 6a, it can be found that the photocurrent is highly dependent on the power density and increases with the power density increasing. This is because the higher the optical power density, the more photons that irradiate to the device surface per unit time, thus, generating more electron–hole pairs. Under the action of the built-in electric field, more electrons are injected into GaN, thus, increasing the current in the device [46]. Figure 6b shows the linear relationship between the photocurrent and the power intensity by fitting with the power law (I ∝ $P^{\theta }$), obtaining a value of θ = 0.97. Here, the value of θ is close to 1, indicating that the Te-enhanced GaN PDs have a high quality contact with electrodes with few traps and defects [47]. Moreover, responsivity (R) and detectivity (D^*^) are two significant figures of the merit for photodetectors. Therefore, the specific responsivity and detectivity of the Te-enhanced GaN PDs illuminated at 365 nm with different power density (bias voltage of 10 V) are quantified in Figure 6c. Here, the responsivity and detectivity are defined by following Equations [48].
(1)$$R=\frac{I_{\mathrm{ph}}-I_{d}}{\mathrm{PS}}$$
(2)$$D^{*}=\frac{A^{1/2}R}{\sqrt{2{\mathrm{qI}}_{d}}}$$
where $I_{\mathrm{ph}}$, $I_{d}$, P, S, A, and q are the photocurrent, dark current, incident optical power density, light absorption area of the photodetector, effective device area, and fundamental unit charge, respectively. (Detailed calculation process are shown in the Appendix A) Here, A equals 3.604 × 10^−5^ cm^−2^. Figure 6c shows that both R and D* increase as the light intensity decreases due to the intensification of carrier complexation at higher light intensities. At the weak power density of 8.49 mW/cm^2^, the maximum R and D* values reach 4951 mA/W and 1.79 × 10^14^ Jones, respectively, which are superior to most reported UV detectors based on GaN, as summarized in Table 1. Figure 6d plots the external quantum efficiency (EQE) as the power density varies by using the following Equation (3) [49]:(3)$$EQE=\frac{Rhc}{e\lambda }$$
where *h* is Planck constant, *c* is the speed of light, *e* is the electronic charge, and λ is the wavelength of the incident light. As calculated, the Te-enhanced GaN PDs can reach a maximum *EQE* of 169%, demonstrating the promising application of this photodetector in faint light detection. The reason that *EQE* > 100% in the optoelectronic devices should be related to the photoconductive gain effect caused by interface trapped states between Te and GaN [50]. Different from the trend of photocurrent variation, the responsivity, detectivity, and *EQE* decrease with the increase in optical power density. This is because when the optical power density increases, the photo-generated carrier concentration increases, weakening the built-in electric field between the Te–GaN interface, resulting in the inhibition of the electron–hole pair separation process and the increase in the electron and hole recombination probability, thus, reducing the responsivity, detectivity, and *EQE* of the device [51,52].

Consequently, to further observe the application of this Te-enhanced GaN PDs as UV photodetectors, the time-resolved measurement is used to judge the photoresponse dynamics. Due to the semimetal property of Te, which can be regarded as a contacted electrode on GaN, the abundant photogenerated carriers in Te that have high electronic conductivity are quickly separated and electrons quickly move to the conduction band of GaN in the forward bias [53]. Figure 6e shows the time-resolved results of the Te-enhanced GaN PDs at a forward bias voltage of 1 V. During the test, the time interval between light on and off was set as 6 s. By analyzing the rise and fall processes of individual response cycles at 1 V (Appendix A), it reveals that the rise and fall times ($\tau_{rise}$ and $\tau_{decay}$) of the Te-enhanced GaN PDs are 100 ms and 270 ms, respectively. These are much faster than the $\tau_{rise}$ = 280 ms and $\tau_{decay}$ = 450 ms of the pure GaN device [54]. This fast response should be the result of a lower energy barrier caused by the formation of a Schottky junction and the high electronic mobility of Te, which implies the great potential of the Te-enhanced GaN PDs for tracking rapidly changing optical signals. To further investigate the influence of bias voltage on the depletion width and the response times of the Te-enhanced GaN PDs, the time-resolved measurement of Te-enhanced GaN PDs with the bias voltage of 5 V and 10 V is conducted. Figure 6f shows the voltage−dependent optical response of GaN and Te-enhanced GaN PDs at 365 nm illumination. The photocurrents of the pure GaN photodetector are 1.12 mA and 3.66 mA, while for Te-enhanced GaN PDs are 1.86 mA and 18.3 mA at bias voltages of 5 V and 10 V, respectively. This significant photocurrent enhancement with the increasing bias voltage is due to the generation of more charge carriers and lower energy barrier with increasing bias voltage. In addition, higher bias voltages enhance the collection of photoexcited carriers, which leads to an increase in photocurrent [55]. In addition, a faster response speed of the Te-enhanced GaN PDs is also observed, compared with bare GaN PDs (Appendix A), which is consistent with the results in Figure 6e.

**Table 1 materials-16-04569-t001:** Comparison of the performance of other photodetectors based on GaN and this work.

Device	λ (nm)	Responsivity (mA/W)	Detectivity (Jones)	Rise/Fall Time	I_on_/I_off_	EQE (%)	Ref.
Te-enhanced GaN	365	4951	1.79 × 10^14^	100/270 ms	2 × 10^3^	169	this work
Te	473	1.6 × 10^5^	−	4.4/2.8 s	1.6	−	[56,57]
GaN	325	340	1.24 × 10^9^	280/450 ms	−	−	[54]
AlGaN	254	3	−	136/15 ms	−	−	[58]
PtSe_2_/GaN	265	193	3.8 × 10^14^	45/102 μs	1 × 10^8^	90.3	[59]
MoS_2_/GaN	265	187	2.34 × 10^13^	46/114 μs	1 × 10^5^	87.5	[60]
BiOCl/ZnO	350	183	−	25.8/11.3 s	798	−	[61]
WS_2_/GaN	375	226	4 × 10^14^	7.3/420 µs	180	74.4	[62]
ZnO/GaN	325	2820	6.82 × 10^13^	6.9/6.4 ms	7.36 × 10^6^	−	[63,64]
MoO_3_/GaN	370	187.5	4.34 × 10^12^	7.55 μs /1.67 ms	1 × 10^5^	62.8	[65]
Te/TiO_2_	350	84	3.7 × 10^9^	0.772/1.492 s	−	−	[66]
Te/ZnO	387	300	4 × 10^10^	2.46/1.75 s	100	−	[66]

## 4. Conclusions

In summary, a simple Te-enhanced GaN-based PD was prepared by using semimetal properties in order to generate energy band bending that can form an internal electric field. The Te-enhanced GaN PDs have outstanding ultraviolet detection performance compared with bare GaN PDs. It is confirmed by the DFT calculations and KPFM tests that the well-matched work function of Te/GaN and metal electrodes provides a special interband tunneling effect, which can greatly suppress the dark current and enhance the photocurrent under ultraviolet irradiation. Benefiting from the above tunneling effect and the fast electronic mobility of Te, Te-enhanced GaN PDs exhibit excellent optical response characteristics at 365 nm illumination, including high responsiveness of 4951 mA/W, ultra-high detectivity of 1.79 × 10^14^ Jones, and an external quantum efficiency of up to 169%. In addition, because Te reduces the contact resistance between GaN and the metal electrodes, the photodetector has a fast response, with a rise time of 100 ms. The photovoltaic performance of Te-enhanced GaN PDs is better than most GaN-based PDs reported in the past. This work promotes the performance of GaN in UV photodetectors and puts forward a new way of developing high-performance photodetectors.

## Figures and Tables

**Figure 1 materials-16-04569-f001:**
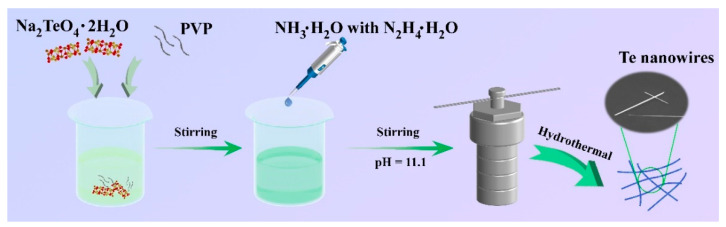
Schematic diagram of the synthesis for Te nanowires.

**Figure 2 materials-16-04569-f002:**
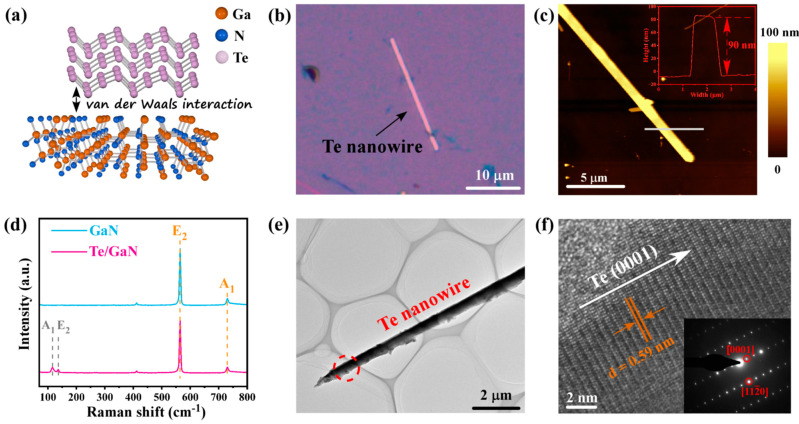
Structure and characterizations of the Te/GaN heterojunction. (**a**) Schematic illustration of the Te/GaN heterojunction. (**b**) Optical image of the Te nanowire. (**c**) AFM image and height plot of the Te nanowire in (**b**). (**d**) Raman characterization of GaN substrate and the heterojunction. (**e**) TEM image of the Te nanowire. (**f**) High-resolution TEM image of the red circular area in (**e**). Inset: SAED patterns of the same area in (**f**).

**Figure 3 materials-16-04569-f003:**
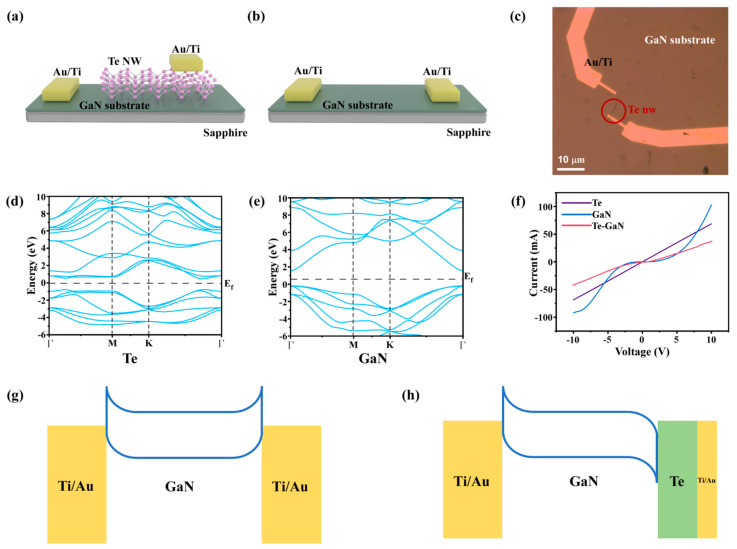
The structure of Te–GaN. (**a**) Schemati diagram of the Te–GaN photodetector. (**b**) Schematic diagram of the bare GaN photodetector. (**c**) Optical microscope image of Te–GaN heterostructure photodetector. (**d**) Electronic structures of Te. (**e**) Electronic structures of GaN. (**f**) I–V curves of the bare Te device, bare GaN device, and the device with Te–GaN. Schematic energy band diagrams of (**g**) bare GaN device and (**h**) Te–GaN device.

**Figure 4 materials-16-04569-f004:**
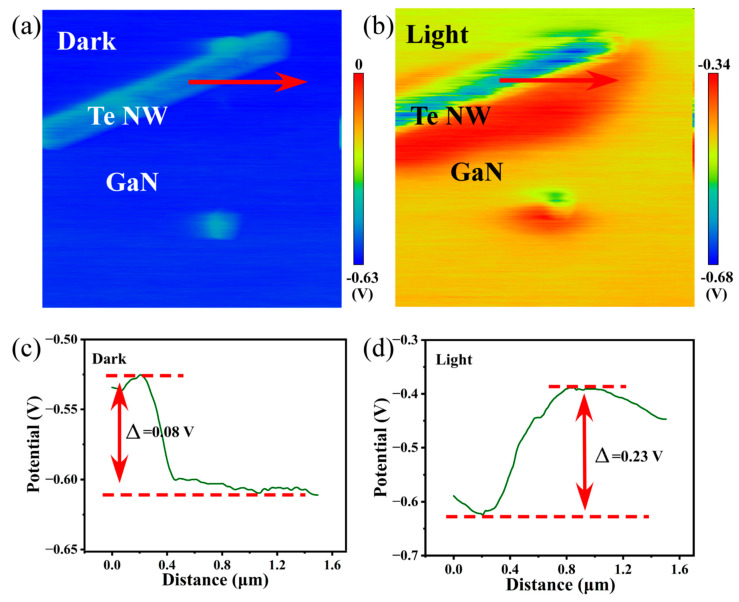
KPFM measurement of Te/GaN heterostructures. (**a**,**b**) Surface potential differences measured in dark and under light illumination (365 nm), respectively. (**c**,**d**) Corresponding potential difference profiles along the lines drawn in (**a**) and (**b**), respectively.

**Figure 5 materials-16-04569-f005:**
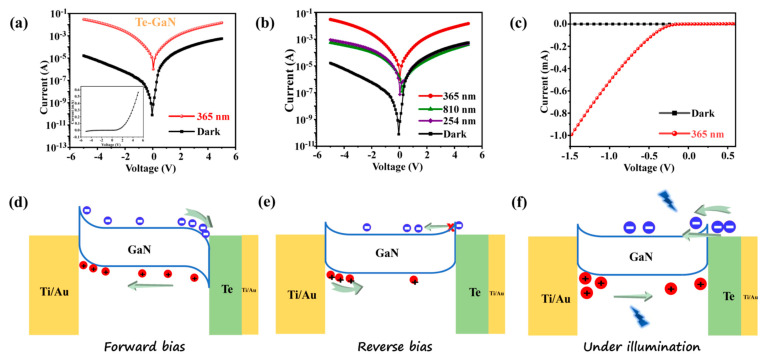
Photovoltaic behavior of Te–GaN PD at different excited wavelength. (**a**) I–V curves of Te–GaN device in the dark and under illumination at 365 nm. (**b**) Photovoltaic behavior of Te–GaN PD at 254 nm, 365 nm, 810 nm, and dark. (**c**) The enlarged I−V curves in the dark and under light of 365 nm. (**d**) Schematic diagram of the bands of Te–GaN interface at forward bias and (**e**) reverse bias. (**f**) Schematic diagram of the bands of Te–GaN interface under 365 nm light irradiation.

**Figure 6 materials-16-04569-f006:**
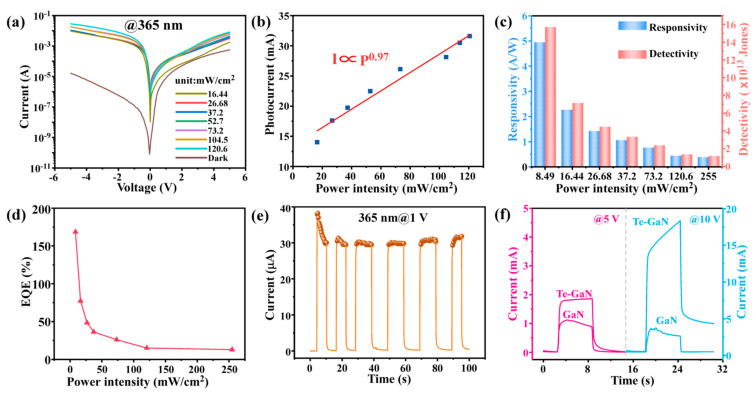
Photovoltaic behavior of the Te-enhanced GaN PDs illuminated at 365 nm with different laser power density. (**a**) The photocurrent of the Te-enhanced GaN PDs as a function of the different illumination intensity measured at the bias voltage from −5 V to 5 V. (**b**) Intensity dependent photocurrent at the bias voltage of 5 V and the corresponding power-law fitting at 365 nm. (**c**) Statistic plot of the responsivity and detectivity of the Te-enhanced GaN PDs at different power intensity. (**d**) Calculated EQE of the Te-enhanced GaN PDs at different power intensity. (**e**) Time-resolved photoresponse of the Te-enhanced GaN PDs, recorded at bias voltage of 1 V and power intensity of 120.6 mw/cm^2^. (**f**) Comparison of the Time-resolved photoresponse of Te–GaN and GaN at different bias voltage of 5 V and 10 V.

## Data Availability

The data presented in this study are available on request from the corresponding author.

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
