# Peer review of "High Performance GaN-Based Ultraviolet Photodetector via Te/Metal Electrodes"

_materials, 2023, doi:10.3390/ma16134569_

Round 1

Reviewer 1 Report (Previous Reviewer 1)

The reviewer thanks the authors for the efforts undertaken for the modification of the manuscript.

Author Response

We thank the positive comments of this reviewer. 

Reviewer 2 Report (Previous Reviewer 3)

I accept this manuscript in its current form. 

Minor Eglish correction is needed before publication.

Author Response

We thank this reviewer for his/her positive comments, and we have revised the writting to correct the manuscripts.

Reviewer 3 Report (New Reviewer)

In this article, the authors have reported 2D material-based Te metal electrodes coated GaN UV photodetectors. The device showed high photoresponse, responsivity, detectivity and EQE, in which the analysis reported values competing with the existing literature. The authors have a solid background in this field. The development theme is interesting and will get a wide readership. As such, we believe that this paper can be worth publishing in the Materials Journal. For further improvement, there are some suggestions in which the referee recommends it to be published after the following revisions.

1)      Picture quality needs to be improved.

2)      In my opinion, r/min would be a little bit confusing for the new reader. I think it would be better if authors could replace it with rpm, otherwise, ignore this comment.

3)      How about the adhesives of Te nanowires? Please explain in the revised manuscript.

4)      Have the authors noticed any shifts in the Raman spectra after the Te was spin-coated over the GaN? Please provide the electrical conductivities of all the as-prepared devices.

5)      In page No: 7, line 1 (…A at ..V), .what does it mean. Please recheck it.

6)      What are the power densities of the particular wavelengths of light used in this research, mainly in Figure 5? What is the distance between the light source and the PD. Were the power densities of each wavelength of light maintained as same or used differently by each? Please provide the wavelength and PD setup details in the experimental section.

7)      Please provide UV-Vis spectra and band gap analysis in order to support the band analysis and DFT band theory.

8)      Is the light absorption area and the incident area of the light illumination is same?

9)      Provide the reason with the appropriate references for the decrease in responsivity, detectivity, and EQE with increase in power densities. Explain elaborately in the revised manuscript. Please go through these papers for your reference 10.1039/c7nr01290j, 10.1002/pssa.202200612, 10.1021/acsanm.2c01410.

10)  What is the importance of power law fitting? Explain detailly. How about the pristine GaN PD? Compare both in supplementary for a better understanding of the effect of Te nanowires on GaN surficial/electronic properties.

11)  Band analysis needs to be improvised, and more interpretations are needed for better insights about your useful research.

Author Response

General Comment:

In this article, the authors have reported 2D material-based Te metal electrodes coated GaN UV photodetectors. The device showed high photoresponse, responsivity, detectivity and EQE, in which the analysis reported values competing with the existing literature. The authors have a solid background in this field. The development theme is interesting and will get a wide readership. As such, we believe that this paper can be worth publishing in the Materials Journal. For further improvement, there are some suggestions in which the referee recommends it to be published after the following revisions.

[Response]

We sincerely thank the reviewer for the thoughtful and encouraging comments about our manuscript and the opportunity to address and clarify the issues raised in the report.  Our responses to the points raised in the report are described below following specific reviewer comments. For the revised parts of the article, we have highlighted them. We believe that these revisions and improvements will make the revised manuscript more reasonable and much stronger, and can be accepted by Materials.

Comment 1-1:

Picture quality needs to be improved.

[Response]

Thanks for the valuable suggestion.We have improved the quality of all the pictures to make the article better.

Comment 1-2:

In my opinion, r/min would be a little bit confusing for the new reader. I think it would be better if authors could replace it with rpm, otherwise, ignore this comment.

[Response]

Thanks for the sincere comment. We have replaced "r/min" with "rpm" in this article to make it more precise and professional.

Comment 1-3:

How about the adhesives of Te nanowires? Please explain in the revised manuscript.

[Response]

Thanks for the kind suggestion. In this system, Te and GaN form 1D/2D heterojunction, and the two materials are bound by van der Waals forces. Therefore, we do not need to use adhesives.

Comment 1-4:

Have the authors noticed any shifts in the Raman spectra after the Te was spincoated over the GaN? Please provide the electrical conductivities of all the as-prepared devices.

[Response]

Thanks for the comment. After Te is coated on GaN, there is almost no shift in the Raman displacement of the two, but the Raman peak intensity of GaN decreases. This result proves that the heterojunction is formed by van der Waals forces. Numerous  conductivity tests were preformed and the representative data is provided in Figure S6 of supplementary information in the form of IV curve.

Comment 1-5:

In page No: 7, line 1 (…A at ..V), .what does it mean. Please recheck it.

[Response]

Thanks for the thoughtful comment. This was a mistake in our work, and we have made this statement complete in page 6, line 29.

Comment 1-6:

What are the power densities of the particular wavelengths of light used in this research, mainly in Figure 5? What is the distance between the light source and the PD. Were the power densities of each wavelength of light maintained as same or used differently by each? Please provide the wavelength and PD setup details in the experimental section.

[Response]

Thanks for the comment. In the test diagram shown in Figure 5, the intensities of the 254 nm, 365 nm, and 810 nm illuminants we used were 33 μW/cm2, 120.6 mW/cm2, and 14.7 mW/cm2, respectively. The distance of laser source and the PDs is 1 m, which can ensure the power density of each light. We have provides the details in the experimental section in page 3, line 29-30.

Comment 1-7:

Please provide UV-Vis spectra and band gap analysis in order to support the band analysis and DFT band theory.

[Response]

Thanks for the kind comment. Since our research objects are based on one-dimensional Te nanowires and two-dimensional GaN films, the UV-VIS spectroscopy test may not suitable for the band gap analysis. In addition, the surface potential and the calculation results of the GaN film and Te have been provided to support the band analysis.

Comment 1-8:

Is the light absorption area and the incident area of the light illumination is same?

[Response]

Thanks for the thoughtful suggestion. In our test, the light incident area is 1 cm2, and the optical absorption area is 3.604×10-5 cm2, which has been mentioned in the article.

Comment 1-9:

Provide the reason with the appropriate references for the decrease in responsivity, detectivity, and EQE with increase in power densities. Explain elaborately in the revised manuscript. Please go through these papers for your reference 10.1039/c7nr01290j, 10.1002/pssa.202200612, 10.1021/acsanm.2c01410.

[Response]

Thanks for the sincere comment. After careful reading of the provided literature , we add the explanation of the decrease in responsivity, detectivity and EQE decrease in page 7, line 36-39 and page 28, line 26-32. And we have cited the referencesas ref. 47, 51, and 53.

Comment 1-10:

What is the importance of power law fitting? Explain detailly. How about the pristine GaN PD? Compare both in supplementary for a better understanding of the effect of Te nanowires on GaN surficial/electronic properties.

[Response]

Thanks for the comment. Through power law fitting, it is found that the relationship between photocurrent and light intensity is very close to linear, but there is a small deviation. This is because trap states caused by defects or impurities in Te or GaN and adsorbed molecules on the Te/GaN interface can capture photogenerated charge, so that absorbed photons cannot be completely converted into photocurrent, resulting in deviation. For the bare GaN PD, we found in the test that its photocurrent did not change with the change of light intensity. Therefore, the relationship between its light intensity and photocurrent has not been studied.

Comment 1-11:

Band analysis needs to be improvised, and more interpretations are needed for better insights about your useful research.

[Response]

Thanks for the comment. We optimize the explanation content of energy band theory and further explain some phenomena.

This manuscript is a resubmission of an earlier submission. The following is a list of the peer review reports and author responses from that submission.

Round 1

Reviewer 1 Report

This paper proposed Te/GaN Type I heterojunctions with enhanced photodetection performance. The fabricated Te/GaN photodetector exhibited better characteristics in terms of dark current, responsivitiy, detectivity, EQE as compared to the typical GaN photodetector. Overall, the paper is interesting and has importance impact on the potential application of GaN based heterojunctions. However, the following issues in this paper have to be addressed for acceptance for publication in Materials journal.

1. In the Materials and Methods section, the procedure of the GaN film grown on sapphire is too short. The controlled growth parameters including gas flow rate, purity of gases/precursors, temperature, pressure, and size of the sapphire substrate should be mentioned.

2. The preparation of the Te NWs, along with Figure 1 should be described in the Materials and Methods section.

3. Details of the fabrication of Te/GaN photodetectors, including lithography and EBL technique are not clearly described in the Materials and Method section.

4. To create the Te/GaN heterojunctions, the Te NWs in ethanol were spin-coated onto the GaN substrate. Any reason why the authors used the spin coating process in this work? Are any other techniques to be proposed to precisely control the deposition of single Te NW prior to the device fabrication?

5. The authors stated that “To deeply investigate the band alignment of the Te/GaN heterojunctions, theoretical studies based on density functional theory (DFT) are conducted on Te, GaN and Te/GaN heterojunctions.” (see Page 4). However, the electronic band structure of Te/GaN heterojunctions is not discussed in the manuscript.

6. The authors stated that “The results show that Te has a direct band gap of 0.17 eV with the Fermi energy level at 5.72 eV, while GaN has an indirect band gap of 1.75 eV with the Fermi energy level at 5.24 eV.” (see Page 4). Any references to support these?

7. The authors stated that “Low work function metals (Ti and Au) are used as the electrodes to ensure the well performance of the devices, such as efficient carrier transport and low interface resistances.” (see Page 5). It is recommended to mention the work function values for Ti and Au.

8. Using the same parameters for the device fabrication of Te/GaN photodetector, any electrical and photoresponse measurements have been conducted for bare GaN photodetector at the wavelength of 365 nm? It is highly recommended to perform the characterization of the GaN photodetector for benchmarking.

9. In the References section, the volume and page number for Ref. 37 are missing. 

Author Response

Review #1

General Comment:

This paper proposed Te/GaN Type I heterojunctions with enhanced photodetection performance. The fabricated Te/GaN photodetector exhibited better characteristics in terms of dark current, responsivity, detectivity, EQE as compared to the typical GaN photodetector. Overall, the paper is interesting and has importance impact on the potential application of GaN based heterojunctions. However, the following issues in this paper have to be addressed for acceptance for publication in Materials journal.

[Response]

We sincerely thank the reviewer for the thoughtful and encouraging comments about our manuscript and the opportunity to address and clarify the issues raised in the report.  Our responses to the points raised in the report are described below following specific reviewer comments. We believe that these revisions and improvements will make the revised manuscript more reasonable and much stronger, and can be accepted by Materials.

Comment 1-1:

In the Materials and Methods section, the procedure of the GaN film grown on sapphire is too short. The controlled growth parameters including gas flow rate, purity of gases/precursors, temperature, pressure, and size of the sapphire substrate should be mentioned.

[Response]

Thanks for the valuable suggestion. The GaN/sapphire substrates were commercial epitaxial GaN wafers on the 2-inch sapphire substrates, which was obtained from Suzhou Nanowin Science and Technology Co.,Ltd. Related content has been modified in the manuscript.

[Revisions]

  • In the main text, page 2, Materials and Methods section, the part, “2.1 Preparation of the GaN Film on Sapphire”, is newly added.

Comment 1-2:

The preparation of the Te NWs, along with Figure 1 should be described in the Materials and Methods section.

[Response]

Thanks for the kind suggestion. The preparation of Te nanowires has been adjusted to the Materials and Methods section.

[Revisions]

  • In the main text, page 2, Materials and Methods section, the part, “2.2 Preparation of the Te nanowires”, is newly added. Figure 1 was moved to page, line.

Comment 1-3:

Details of the fabrication of Te/GaN photodetectors, including lithography and EBL technique are not clearly described in the Materials and Method section.

[Response]

Thanks for the thoughtful comment. We have rewritten the Materials and Method section to provide the details about the fabrication of devices.

[Revisions]

  • In the main text, page , Materials and Methods section, the part, “2.5 Fabrication of Te/GaN photodetectors”, is revised to add details about the fabrication process.

Comment 1-4:

To create the Te/GaN heterojunctions, the Te NWs in ethanol were spin-coated onto the GaN substrate. Any reason why the authors used the spin coating process in this work? Are any other techniques to be proposed to precisely control the deposition of single Te NW prior to the device fabrication?

[Response]

Thanks for the sincere comment. This work used spin coating process is due to that it can provide an efficient and convenient approach to get the heterojunctions with controllable thickness. Although CVD is another widely used technique to obtain the heterojunctions recently, the thickness and distribution of the materials is difficult to be controlled. Besides, to precisely control the deposition of single Te NWs, it usually needs to locate the deposition of Te seeds in advance, which should be achieved by a series of cumbersome procedures, such as lithography and EBL technique.

Comment 1-5:

The authors stated that “To deeply investigate the band alignment of the Te/GaN heterojunctions, theoretical studies based on density functional theory (DFT) are conducted on Te, GaN and Te/GaN heterojunctions.” (see Page 4). However, the electronic band structure of Te/GaN heterojunctions is not discussed in the manuscript.

[Response]

Thanks for the thoughtful suggestion. Here, we provide the experimental calculations for the bandgap of the Te/GaN junction. Based on the result, it shows that due to the narrow bandgap and the semimetal property of Te, the whole junction has the metallic property at the interface, which proves that by contacted with Te, the mobility of the photogenerated carriers in Te/GaN can be greatly improved.

[Revisions]

(1)In the Supplementary material,we have added the electronic structures of Te/GaN heterojunction in figure S2.

Comment 1-6:

The authors stated that “The results show that Te has a direct band gap of 0.17 eV with the Fermi energy level at 5.72 eV, while GaN has an indirect band gap of 1.75 eV with the Fermi energy level at 5.24 eV.” (see Page 4). Any references to support these?

[Response]

Thanks for the comment. We have provided the related references about the band gap of Te and GaN in the manuscript to support our results. Also, the related references have been added into the corresponding discussions.

[Revisions]

(1) In the main text, page 5 , line 7, two related references have been newly added as No.38 and No.39.

Comment 1-7:

The authors stated that “Low work function metals (Ti and Au) are used as the electrodes to ensure the well performance of the devices, such as efficient carrier transport and low interface resistances.” (see Page 5). It is recommended to mention the work function values for Ti and Au.

[Response]

Thanks for the outstanding comment. Following your suggestion, we have detailed the work functions of Ti and Au in the article and added the advantages of choosing Ti/Au as electrodes.

[Revisions]

  • In the main text, page 5, line 23, we add the contents of “In addition, the work functions of Au and Ti (5.1 eV, 4.33 eV) are close to those of Te and GaN (4.95 eV, 4.1 eV), When Ti/Au is chosen as the electrode material for the device, the contact barrier between the semiconductor layer and the source and drain can be decreased to obtain the better electrical performance.”

Comment 1-8:

Using the same parameters for the device fabrication of Te/GaN photodetector, any electrical and photoresponse measurements have been conducted for bare GaN photodetector at the wavelength of 365 nm? It is highly recommended to perform the characterization of the GaN photodetector for benchmarking.

[Response]

Thanks for the valuable comment. We have fabricated the devices of bare GaN to test the electrical and photoresponse properties at 254 nm and 365 nm (Figure R1). Besides, the the time-resolved photoresponse of bare GaN at 5 V and 10 V under the illumination of 365 nm (120.6 mW/cm2) was also provided for comparison in Figure 5f.

Figure R1 Photovoltaic behavior of GaN at 254 nm and 365 nm.

[Revisions]

  • In the Supplementary material,we have added the Photovoltaic behavior of GaN in figure S4.

Comment 1-9:

In the References section, the volume and page number for Ref. 37 are missing. 

[Response]

Thanks for the careful examination. This was a mistake in our work and we feel guilty about it. We have amended this section in the article.

[Revisions]

  • In the references section, we have changed the 37 to “Liu X.;Zhou J. ReS2 on GaN Photodetector Using H+ Ion-Cut Technology. ACS omega 2023, 8, 457-463.”

Reviewer 2 Report

This study looks interesting for publishing. However, some interesting points that should b considered are below:

-          Fig 4 (c), I am a bit interested in the behavior of the photocurrent is very high in the negative voltage range, however, it is almost zero in the positive part. Is it a problem with electrode connections? Or maybe Au/Ti is not effective in extractive the photocurrents well?

-          I am a little bit scared about the electrical connections between Te and Au/Ti

-          Did your measurements carry out only for a single nanowire? Or multiple?

-          Can you put the experimental calculations of the bandgap of the combined device?

-          Why did you use a very high 10 V bias voltage? People now are more interested in self-powered photodetectors or at least low-bias voltages. what do you think? can you report the detetctivity of your device at 0V?

- In Fig. S2, it is expected that the Te itself is metallic right, how can you get this IV? How did you get the measurements here from the Te only? is deposited on another substrate? Can you explain it

-          Can you merge Fig. S3. And S2 better? Then please put it in the main manuscript because it is important  

Author Response

General Comment:

This study looks interesting for publishing. However, some interesting points that should be considered are below:

[Response]

Thanks for the kind comment. Based on these comments, we have made responsive changes to the article to improve it.

Comment 2-1:

Fig 4 (c), I am a bit interested in the behavior of the photocurrent is very high in the negative voltage range, however, it is almost zero in the positive part. Is it a problem with electrode connections? Or maybe Au/Ti is not effective in extractive the photocurrents well?

[Response]

Thanks for the valuable comment. Actually, this is the typical rectification behavior of p-n diodes, which is also suitable for our Te (p type)/GaN (n type) heterojunctions. In the manuscript, we have explained that when reverse bias is applied to the device at 365 nm, lots of photos generated electrons in Te can overcome the steep energy barrier and jump to the conduction band of GaN due to the photon energy of 365 nm is much higher than the band gap of Te. As a result, electrons can easily drift from Te to GaN under the reverse bias (Figure 4f). In contrast, when applying forward bias to the device, the light does not have such a facilitating effect on the photocurrent.

Comment 2-2:

 I am a little bit scared about the electrical connections between Te and Au/Ti

[Response]

Thanks for the thoughtful comment. The reason that we used Au/Ti as the electrodes has been added in the manuscript. Besides, the I-V curves of bare Te devices show that the contact between Te and Au/Ti is quiet well, as Figure 3f. We also provide more results on the devices of Te nanoplates/GaN in the Supporting Information to show the well contact between Te and GaN.

[Revisions]

  • In the main text, page 5, line 23, we add the contents of “In addition, the work functions of Au and Ti (5.1 eV, 4.33 eV) are close to those of Te and GaN (4.95 eV, 4.1 eV), When Ti/Au is chosen as the electrode material for the device, the contact barrier between the semiconductor layer and the source and drain can be decreased to obtain the better electrical performance.”
  • In the Supplementary material,we have added the device of Te nanoplates/GaN in figure S6.

Comment 2-3:

Did your measurements carry out only for a single nanowire? Or multiple?

[Response]

Thanks for the valuable comment. All the measurements of electrical and photovoltaic properties were carried out on a considerable number of Te/GaN devices to obtain the convincible results. Besides, we have provided more optical images of the devices in Figure S5.

[Revisions]

  • In the main text, page, line, one sentence, “Here, plenty of devices have been fabricated to obtain the convincible results (Figure S4).”, has been newly added.
  • In the Supporting Information, Figure S4 is newly added.

Comment 2-4:

Can you put the experimental calculations of the bandgap of the combined device?

[Response]

Thanks for the thoughtful suggestion. Here, we provide the experimental calculations for the bandgap of the Te/GaN junction. Based on the result, it shows that due to the narrow bandgap and the semimetal property of Te, the whole junction has the metallic property at the interface, which proves that by contacted with Te, the mobility of the photogenerated carriers in Te/GaN can be greatly improved.

[Revisions]

(1)In the Supplementary material,we have added the electronic structures of Te/GaN heterojunction in figure S2.

Comment 2-5:

Why did you use a very high 10 V bias voltage? People now are more interested in self-powered photodetectors or at least low-bias voltages. what do you think? can you report the detectivity of your device at 0V?

[Response]

Thanks for the valuable comment. We agree that self-powered photodetectors and photodetectors with low-bias voltages are more promising recently. In this work, we also get the considerable photocurrent of 2.71 mA at -1.5 V, which is much larger than the bare GaN (1.03 mA). More importantly, based on our results of the photovoltaic behavior under the illumination with different wavelength (Figure 4b), the Te/GaN heterojunctions can also be used as the broadband photodetectors. Our work is mainly focus on the promotion of the photovoltaic behavior of GaN, and lacks of the consideration of bias voltage in working condition. Unfortunately, the current devices only have the small photocurrent of 19 μA at 0 V. To solve this weakness, further studies will be conducted on the structure design of the devices to achieve the self-power performance.

Comment 2-6:

In Fig. S2, it is expected that the Te itself is metallic right, how can you get this IV? How did you get the measurements here from the Te only? is deposited on another substrate? Can you explain it

[Response]

Thanks for the kind comments. For the IV test, we used the same process to fabricate the device of bare Te NWs on SiO2. Also, Te NWs were transferred onto the substrates by the spin coating process. Then, the four-point probe measurement was conducted to test the electrical behavior of Te.

[Revisions]

  • In the main text, page 5, line 27, one sentence, “For the test of bare Te NWs and GaN substrate, the four-point probe measurement was conducted to obtain the electronical behavior.”, has been newly added.

Comment 2-7:

Can you merge Fig. S3. And S2 better? Then please put it in the main manuscript because it is important  

[Response]

Thanks for the valuable comment. We have merged Figure S2 and S3 into one figure, shown as Figure 3f, in the main text according to the suggestion.

[Revisions]

(1)In the main text, page 5, figure 3, we have added the related contents in figure 3f, and adjusted the orders of figure 3.

Reviewer 3 Report

In this manuscript, authors made Te/GaN type-I heterojunction with the straddling band structure on the sapphire substrate. Here authors investigated the tunneling effect and the ultrahigh electronic mobility of Te NWs which effectively collected the photogenerated carriers under illumination. Thus, the PDs based on Te/GaN heterojunction showed a promising responsivity of 4951 mA/W, the detectivity of 1.57 × 1016 Jones and the external quantum efficiency of 169%. This work looks very interesting, but it needs few issues to resolved before publication as:

1.      In this sentence of introduction “which usually feature the advantages of high responsivity and “on/off” ratio under applied bias voltage”.  The ‘on/off’ of which factor?

2.      In Figure 2b, how authors controlled the concentration of Te wires. Does it seem that in 130~ 150 um of area they found only one Te wire?

3.      In Figure 2c, the width of Te wire looks 400 nm (0.4 um) but in Figure 2c of the optical image it looks more the 400 nm. Also, provide the real image of AFM.

4.      In Figure 3d, the device is constructed line Au/Te/GaN/Au. Please mention the source and drain electrodes clearly.

5.      Importantly, in Figure 3d it seems that both contacts are connected with GaN. But it must be like: one is connected with GaN and the other should be only connected with Te wire to make proper Au/Te/GaN/Au configuration to avoid short circuit issue. Please explain this categorically.

6.      In equation 1, S is the area. So, what is an area of the device here?

7.      How authors calculated the rise and decay time. Please mention the formula or equation in the main manuscript. Also, provide each detail (fitting and error) of time calculation in the response file.

8.      From where the authors extracted the data of Figure 5b?

9.      What is the illumination of one cycle in Figure 5e and 5f?

10.   In Figure S2 and S3, if we look at IV curves of Au/GaN/Au and Au/Te/GaN/Au which seems identical. Please give a reason for this issue. To me, it shows a short circuit.

11.   Also, for broad readership of this manuscript please add a few more articles in first paragraph of introduction. DOI: 10.1039/D0DT01164A and https://doi.org/10.1039/C7TC04754A

Author Response

General Comment:

In this manuscript, authors made Te/GaN type-I heterojunction with the straddling band structure on the sapphire substrate. Here authors investigated the tunneling effect and the ultrahigh electronic mobility of Te NWs which effectively collected the photogenerated carriers under illumination. Thus, the PDs based on Te/GaN heterojunction showed a promising responsivity of 4951 mA/W, the detectivity of 1.57 × 1016 Jones and the external quantum efficiency of 169%. This work looks very interesting, but it needs few issues to resolved before publication as:

[Response]

We sincerely appreciate for the inspiring comments about our manuscript. Our responses to the points raised in the report are described below following specific reviewer’s comments.

Comment 3-1:

In this sentence of introduction “which usually feature the advantages of high responsivity and “on/off” ratio under applied bias voltage”.  The ‘on/off’ of which factor?

[Response]

Thanks for the valuable suggestion. The "on/off" generally means that the photocurrent of the devices at the “on” and “off” state of the light source, which can be obtained by conducting transient response test or spectral response test on the device. This factor can reflect the current control ability of the device.

Comment 3-2:

In Figure 2b, how authors controlled the concentration of Te wires. Does it seem that in 130~ 150 um of area they found only one Te wire?

[Response]

Thanks for the kind comment. The Te nanowires were synthesized by a hydrothermal method. During the spin coating process, the Te NWs obtained by hydrothermal were dispersed in ethanol after sonication to obtain the dispersion. The concentration of Te nanowires can be controlled by regulating the concentration of Te NWs. We made several attempts to adjust the density of tellurium NWs to facilitate our manipulation of the devices.

[Revisions]

  • In the main text, page, line, one sentence, “The density of Te NWs on the substrate can be adjusted by the concentration of Te NWs in the dispersion.”, has been newly added.

Comment 3-3:

In Figure 2c, the width of Te wire looks 400 nm (0.4 um) but in Figure 2c of the optical image it looks more the 400 nm. Also, provide the real image of AFM.

[Response]

Thanks for the thoughtful comment. We have changed the optical image and the corresponding AFM image in Figure 2b and 2c.

[Revisions]

  • In the main text, page4, Figure 2b and 2c have been changed.

Comment 3-4:

In Figure 3d, the device is constructed line Au/Te/GaN/Au. Please mention the source and drain electrodes clearly.

[Response]

Thanks for the sincere comment. Based on your suggestion, we have revised the manuscript.

[Revisions]

(1) In the main text, page 4, line 35, We have added the description of the source and drain contents as “During device testing, Te is the drain electrode and GaN is the source electrode.”

Comment 3-5:

Importantly, in Figure 3d it seems that both contacts are connected with GaN. But it must be like: one is connected with GaN and the other should be only connected with Te wire to make proper Au/Te/GaN/Au configuration to avoid short circuit issue. Please explain this categorically.

[Response]

Thanks for the suggestion. Here, the electronic conductivity of Te is much better than that of GaN, which performs almost like a semimetal material (Figure 3). Therefore, electrons will follow from Au/Ti to GaN, and then to the Te/Ti/Au electrode. The role of Te is to adjust the work function between GaN and Ti/Au and to accelerate the mobility of photogenerated carriers in GaN, proved by the photovoltaic results. We may define this device as a GaN based PD improved by Te/GaN Schottky junction, similar to the previous reports (Photon. Res. 2021, 9, 734-740 and Opt. Lett. 2022, 47, 1561-1564). Based on this, we have made some modification in the discussion to make a clear interpretation.

[Revisions]

  • In the main text, page 5, line 8, we have changed the energy band interpretation of the interface between Te/GaN and metal electrode.
  • In the main text, page 6, line 19, before discussing the effect of light on the energy band, we emphasize the role of Te as a contact metal to promote the photocurrent of GAN-based photodetectors.

Comment 3-6:

In equation 1, S is the area. So, what is an area of the device here?

[Response]

Thanks for the outstanding comment. We have recalculated the device area by the area between source and drain electrodes.And the value of area is 3.604×10-5 cm-2.

Comment 3-7:

How authors calculated the rise and decay time. Please mention the formula or equation in the main manuscript. Also, provide each detail (fitting and error) of time calculation in the response file.

[Response]

Thanks for the kind suggestion. The rise time is defined as the time interval for the response to rise from 10% to 90% of its peak value, whereas the fall time is defined as the time interval for the response to decay from 90% to 10% of its peak value. In the calculation, we selected the parts of the rising and falling curve respectively for linear fitting, and then calculated the response time according to the fitting function. The relevant calculation procedure and fitting data have been added to the Supplementary Information.

Comment 3-8:

From where the authors extracted the data of Figure 5b?

[Response]

Thanks for the valuable comment. Figure 5b shows the photocurrent values corresponding to different light intensity under the bias voltage of 5 V selected in figure 5a, and the results are obtained by fitting according to the power function I  .

Comment 3-9:

What is the illumination of one cycle in Figure 5e and 5f?

[Response]

Thanks for the comment. Figure 5e and 5f show the transient photocurrent response of the device. During the test, the time interval between light (illuminated at 365 nm) on and off was set as 6 s, in which the optical power density was 120 mW/cm2, and the test result was the image shown in the article.

[Revisions]

(1)In the main text,page 8,line 7, relevant information has been added.

Comment 3-10:

In Figure S2 and S3, if we look at IV curves of Au/GaN/Au and Au/Te/GaN/Au which seems identical. Please give a reason for this issue. To me, it shows a short circuit.

[Response]

Thanks for the thoughtful comment. The IV curves of GaN and Te/GaN heterojunction are totally different. In order to present them in a clear way, we reorganize the results, shown as Figure 3f, in the main text.

Figure R1. I-V curves of the bare Te device, bare GaN device and the device with Te/GaN junction.

Comment 3-11:

Also, for broad readership of this manuscript please add a few more articles in first paragraph of introduction. 

DOI: 10.1039/D0DT01164A and https://doi.org/10.1039/C7TC04754A

[Response]

Thanks for the valuable comment. We have enriched this manuscript by incorporating more references according to the comments.

[Revisions]

  • In the main text, page , line, two more references have been newly added as No.9 and No.10.

Round 2

Reviewer 2 Report

Thank you for your report 

Author Response

Dear reviewer,
Thank you for your approval of this article. We have further optimized the statements and expressions in the article.Thanks again for your interest in this article.

Reviewer 3 Report

The authors have tried well to reply to the comments but several more questions were raised after revision as given below:

1.      The quality of all figures is very low, Figure 2c and Figure (3e, 3f) are especially invisible. The quality of Figure was quite well in the first manuscript.

2.      It is hard to make the decision about thickness of Te wire with given AF image.

3.      I have serious concern on device structure to claim Te/GaN heterojunction. When you need to fabricate the heterojunction of Te/GaN then one Au contact is completely connected with only GaN and the other Au contact is completely connected with Te wire only. But in this manuscript the electrode which should be connected only with Te is also touched with GaN. So, this kind of structure does not represent the Te/GaN heterojunction but only Au/GaN/Au configuration. So, authors should think about it carefully. For more convenient please read this article very carefully.  (DOI 10.1088/1361-6528/aa9eb8)

4.      Also, the first sentence of introduction should be more concise and comprehended. Please rewrite the 1st sentence and cite these recent articles:

(A)  doi: https://doi.org/10.1016/j.apcatb.2022.121363

(B)   https://doi.org/10.1002/adom.202201396

(C)  Encapsulation strategies on 2D materials for field effect transistors and photodetectors. CHINESE CHEMICAL LETTERS, 2022. 33(5): p. 2281-2290

5.      Further, authors discussed the physiochemical properties on page 1 line 39-42. So, I should recommend to make a small paragraph there to explain it more relevant with photoelectrochemical sensor and perovskite films for sensitive photodetectors. Please add these references:

(A)  Label-free photoelectrochemical sensor based on 2D/2D ZnIn2S4/g-C3N4 heterojunction for the efficient and sensitive detection of bisphenol A. CHINESE CHEMICAL LETTERS, 2022. 33(2): p. 983-986.

(B)  Self-powered anti-interference photoelectrochemical immunosensor based on Au/ZIS/CIS heterojunction photocathode with zwitterionic peptide anchoring. CHINESE CHEMICAL LETTERS, 2022. 33(11): p. 4750-4755.

(C)  Chemical vapor deposition growth of phase-selective inorganic lead halide perovskite films for sensitive photodetectors. CHINESE CHEMICAL LETTERS, 2021. 32(1): p. 489-492.

Author Response

Dear reviewer,

We are very honored to receive your professional comments. This is our reply after careful consideration. The specific content is as follows:

Comment 3-1:

The quality of all figures is very low, Figure 2c and Figure (3e, 3f) are especially invisible. The quality of Figure was quite well in the first manuscript.

[Response]

Thanks for the valuable suggestion. We have improved the quality of all figures. For figures 3, we have resized them to make these figures more clear and crisp.

Comment 3-2:

 It is hard to make the decision about thickness of Te wire with given AF image.

[Response]

Thanks for the kind suggestion. To make it easier to see the AFM results, we have added scales and enlarged the illustrations and values.

Comment 3-3:

   I have serious concern on device structure to claim Te/GaN heterojunction. When you need to fabricate the heterojunction of Te/GaN then one Au contact is completely connected with only GaN and the other Au contact is completely connected with Te wire only. But in this manuscript the electrode which should be connected only with Te is also touched with GaN. So, this kind of structure does not represent the Te/GaN heterojunction but only Au/GaN/Au configuration. So, authors should think about it carefully. For more convenient please read this article very carefully.  (DOI 10.1088/1361-6528/aa9eb8)

[Response]

Thanks for the thoughtful comment. Based on the advice, we carefully revised the whole manuscript to define the device as the Te-enhanced GaN PDs. Here, Te is used as a contact material between the metal electrodes and GaN, which can greatly reduce the contact resistance and solve the Fermi pinning effect. In the manuscript, we change the original Te/GaN heterojunction device definition to Te-enhanced GaN based photodetector. The relevant discussions have also been changed, together with the title.

Comment 3-4:

Also, the first sentence of introduction should be more concise and comprehended. Please rewrite the 1st sentence and cite these recent articles:

(A)  doi: https://doi.org/10.1016/j.apcatb.2022.121363

(B)   https://doi.org/10.1002/adom.202201396

(C)  Encapsulation strategies on 2D materials for field effect transistors and photodetectors. CHINESE CHEMICAL LETTERS, 2022. 33(5): p. 2281-2290

[Response]

Thanks for the comment.We have made some changes to the introduction to make it more concise, and we have referenced the relevant literature mentioned in the manuscript.

[Revisions]

  • In the main text, page1, line 42, three more references have been newly added as No.10, No.11, and No.12.

Comment 3-5:

 Further, authors discussed the physiochemical properties on page 1 line 39-42. So, I should recommend to make a small paragraph there to explain it more relevant with photoelectrochemical sensor and perovskite films for sensitive photodetectors. Please add these references:

(A)  Label-free photoelectrochemical sensor based on 2D/2D ZnIn2S4/g-C3N4 heterojunction for the efficient and sensitive detection of bisphenol A. CHINESE CHEMICAL LETTERS, 2022. 33(2): p. 983-986.

(B)  Self-powered anti-interference photoelectrochemical immunosensor based on Au/ZIS/CIS heterojunction photocathode with zwitterionic peptide anchoring. CHINESE CHEMICAL LETTERS, 2022. 33(11): p. 4750-4755.

(C)  Chemical vapor deposition growth of phase-selective inorganic lead halide perovskite films for sensitive photodetectors. CHINESE CHEMICAL LETTERS, 2021. 32(1): p. 489-492.

[Response]

Thanks for the comment.We have explained and supplemented the relationship between the perovskite thin film and the photodetector, and enriched this manuscript by incorporating more references according to the comments.

[Revisions]

  • In the main text, page1, line 42-46, we added some content related to perovskite and photodetectors.
  • In the main text, page1, line 46, three more references have been newly added as No.13, No.14, and No.15.
